# Transoral Marsupialization of an Isolated Surgical Ciliated Cyst of the Infratemporal Fossa

**DOI:** 10.3390/diagnostics13111825

**Published:** 2023-05-23

**Authors:** Da-Woon Kwack, Jooha Yoon, Hae-Seo Park, Jung-Hwan Lee, Moon-Young Kim

**Affiliations:** 1Department of Oral and Maxillofacial Surgery, College of Dentistry, Dankook University, Cheonan 31116, Chungcheongnam-do, Republic of Korea; butterfluu@gmail.com (D.-W.K.); jooha0123@naver.com (J.Y.); 12120509@dankook.ac.kr (H.-S.P.); 2Institute of Tissue Regeneration Engineering (ITREN), Dankook University, Cheonan 31116, Chungcheognam-do, Republic of Korea; 3UCL Eastman-Korea Dental Medicine Innovation Centre, Dankook University, Cheonan 31116, Chungcheognam-do, Republic of Korea; 4Department of Nanobiomedical Science and BK21 PLUS NBM Global Research Center for Regenerative Medicine, Dankook University, Cheonan 31116, Chungcheognam-do, Republic of Korea; 5Department of Biomaterials Science, College of Dentistry, Dankook University, Cheonan 31116, Chungcheognam-do, Republic of Korea; 6Cell & Matter Institute, Dankook University, Cheonan 31116, Chungcheongnam-do, Republic of Korea

**Keywords:** nonodontogenic cysts, post-traumatic, skull base, infratemporal fossa, minimally invasive surgical procedures

## Abstract

Surgical ciliated cysts occur primarily in the maxilla after radical maxillary sinus surgery. We report the first case of a surgical ciliated cyst that developed in the infratemporal fossa 25 years after the patient sustained severe facial trauma. The patient complained of mandibular pain and limited mouth opening. The patient’s condition was completely resolved 5 months after marsupialization via Le Fort I osteotomy. Surgical morbidities can be minimized by proper diagnosis and less invasive surgery.

Surgical ciliated cysts, also known as postoperative maxillary cysts, respiratory implantation cysts, or paranasal cysts, are the collective term for non-odontogenic cysts lined by the respiratory epithelium because of traumatic implantation to the sinus or nasal mucosa [1]. It occurs most commonly in the maxilla, whereas it rarely occurs in the mandible due to implantation in the sinus epithelium by contaminated instruments or due to using nasal bone or cartilage with the epithelium for augmentation genioplasty [2,3]. Although this condition has been reported, its occurrence in the infratemporal fossa has not yet been reported.

Here, we present a case of a surgical ciliated cyst in the infratemporal fossa that was completely resolved using marsupialization via Le Fort I osteotomy.

A 40-year-old man was referred to the Department of Oral and Maxillofacial Surgery with mandibular pain and limited mouth opening. Symptoms had started a year previously and gradually deteriorated. He was otherwise healthy but had experienced a facial bone fracture 25 years ago. On initial clinical examination, the maximum mouth opening was 26 mm because of pain. Radiographic investigations included Panorex imaging (Figure 1a), a transcranial view of the temporomandibular joint (Figure 1b), computed tomography (CT) with contrast (Figure 2a,b), and magnetic resonance imaging (MRI) (Figure 2c,d). The studies confirmed a well-defined, large osteolytic lesion 2.8 × 5.2 × 3.8 cm in the right infratemporal fossa with extension into the maxillary sinus and foramen ovale.

Under general anesthesia, old plates and screws on the anterior maxilla were removed, and a standard Le Fort I osteotomy was performed. After the downfracture, the lesion was carefully dissected and visualized. Aspiration was performed to confirm the lesion was a cyst, and slightly yellow, serous fluid was withdrawn. A partial excision of an ovoid shape was made in the cystic wall (0.8 × 2.4 × 0.3 cm) for a histopathologic exam. After repositioning the maxilla, a 1 × 1 cm sized bony window was made in the canine fossa area. The 10 cm Penrose drain tube was inserted at the end of the cyst through the anterior bony window. The drain was secured in place with a 3-0 non-absorbable suture to the anterior vestibular mucosa. The operative course was uneventful, and the patient was discharged on the third postoperative day. Before discharge, the patient was instructed to independently irrigate the wound with saline using a syringe after each meal. Histopathologic evaluation showed a pseudostratified ciliated columnar epithelium and focally squamous epithelium compatible with the diagnosis of a surgical ciliated cyst (Figure 3).

Regular follow-up visits were scheduled every other month for Penrose drain adjustments. The drain was shortened by 1 cm monthly. A follow-up CT scan performed 2 months after surgery showed a decreasing bony defect and regenerating bone around the drain. The drain was fully removed 5 months after surgery. A CT scan taken 3 years postoperatively showed evidence of complete bony regeneration without recurrence (Figure 4a,b).

The infratemporal fossa is a rectangular posterior maxillary space located deep in the skull base that contains important neurovascular structures such as the pterygoid muscle, mandibular nerve, tympanic cord, middle meningeal artery, internal maxillary artery and vein, and pterygoid venous plexus [4]. The foramen ovale is just posterior to the junction of the lateral pterygoid plate with a sphenoid body, and the foramen spinosum is located a few millimeters posterolaterally [5]. The classic approaches to the infratemporal fossa can be categorized into anterior, lateral, and transcranial pathways [6]. Because of the significant morbidities associated with invasive surgery of the infratemporal fossa, less invasive alternatives are recommended [7].

Le Fort I osteotomy was neglected as a surgical approach for decades before regaining popularity, especially for the removal of tumors at the base of the skull and midface, in the late 1980s [8,9]. The major advantage of this procedure is that it provides a wide surgical field of view in the maxillary sinus and nasopharynx, allowing large cysts such as those described in this article to be easily exposed without needing an otherwise extended and unacceptable resection of the maxillary bone. Disadvantages are related to the classic intra- and perioperative complications associated with maxillary osteotomies, such as anatomic deviation, hemorrhage, infection, malocclusion, nonunion or pseudoarthrosis, maxillary sinusitis, and maxillary bone necrosis, which are rare [10].

For the treatment of surgical ciliated cysts, most previous studies have recommended a Caldwell–Luc operation consisting of the complete removal of the cyst lining, together with nasal antrostomy [11,12,13]. Primary closure or open packing yielded equally effective results, and marsupialization may be indicated for the treatment of unilocular cysts with a thin cystic wall and extensive bony perforation or when cysts are located in an inaccessible location [14].

In our patient, based on the high T1/T2 signal detected via MRI, the lesion was suspected to be cystic. Some benign tumors, such as epidermal inclusion cysts, fibromas, and schwannomas may present with similar features in the infratemporal fossa. Differential diagnosis includes neurogenic tumors, lymphangioma dermoid/epidermoid cysts, and surgical ciliated cysts. Because the lesion was regarded as non-malignant, we considered surgical excision immediately without performing an incisional biopsy before surgery. During surgery, complete enucleation of the lesion without injury to the adjacent structures was considered impossible because of the thin cystic wall and extensive bone loss, and the lesion was marsupialized.

Surgical ciliated cysts occur as a delayed complication after radical surgical intervention and trauma in the maxillary sinus and rarely recur after appropriate surgical treatment. Surgical procedures are determined according to the location or aggressiveness of the lesion. This is the first report of the marsupialization via Le Fort I osteotomy of a surgical ciliated cyst in the deep infratemporal fossa. The risk of significant morbidity associated with an isolated surgical ciliated cyst of the infratemporal fossa can be minimized by proper diagnosis and less invasive surgery, including the most appropriate surgical approach.

## Figures and Tables

**Figure 1 diagnostics-13-01825-f001:**
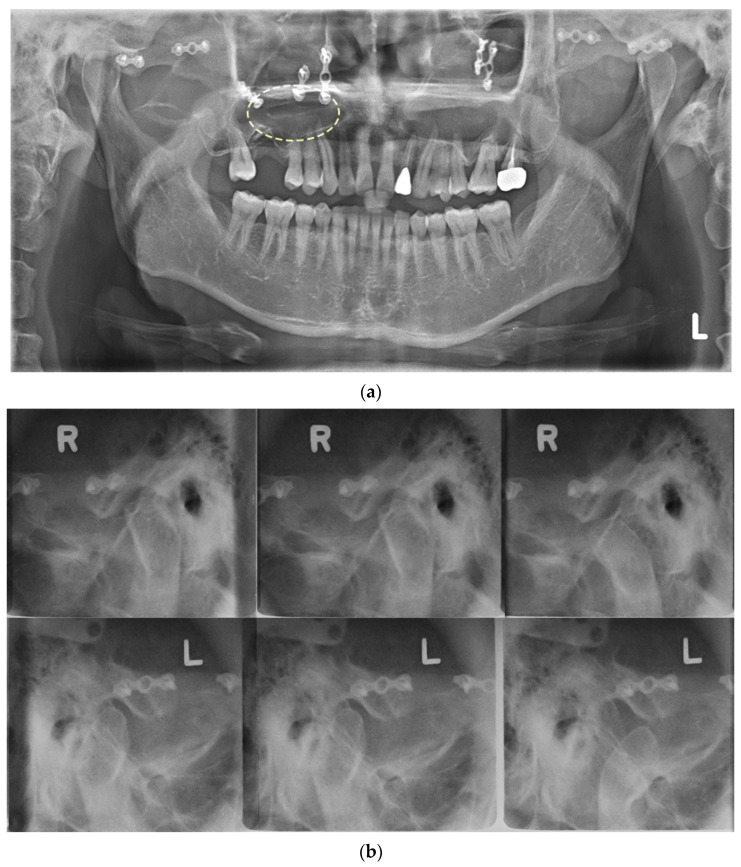
Plain radiographs taken at first visit. (**a**) Panorex showing old miniplates and screws on the midface area (dash circle); (**b**) in the transcranial view of the temporomandibular joint, no significant bony changes of the condyles were observed, but the movement of both condyles appeared to be restricted.

**Figure 2 diagnostics-13-01825-f002:**
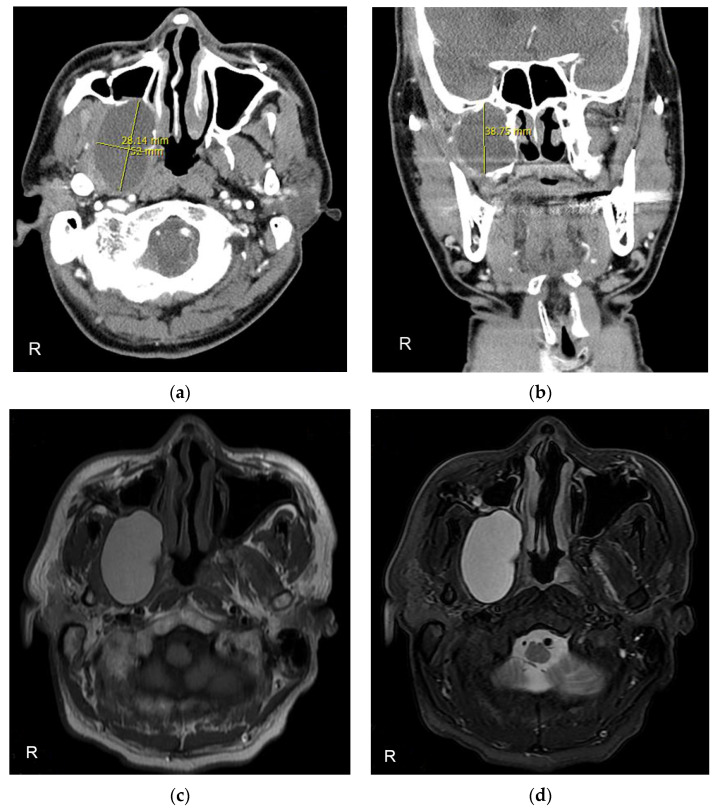
Preoperative computed tomography (CT) with contrast and magnetic resonance imaging (MRI) scans showing a well-defined osteolytic 2.8 × 5.2 × 3.8 cm lesion extended into the right maxillary sinus and pterygoid plate. (**a**) Axial view of CT. (**b**) Coronal view of CT. (**c**) Axial view of T1-weighted MRI. (**d**) Axial view of T2-weighted MRI.

**Figure 3 diagnostics-13-01825-f003:**
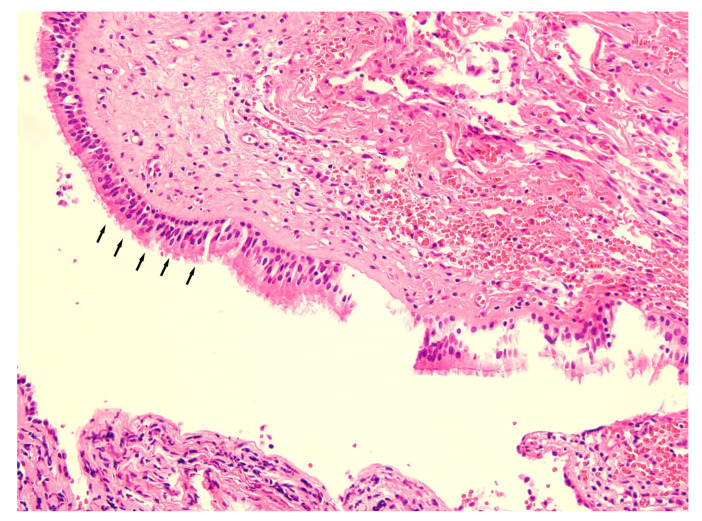
Histopathological features of pseudostratified epithelial lining (arrow) with papillary arrangement (hematoxylin and eosin, original magnification ×200).

**Figure 4 diagnostics-13-01825-f004:**
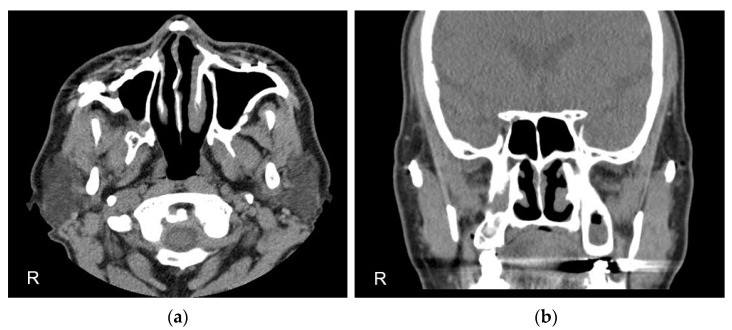
CT scan 3 years after surgery showing complete resolution without recurrence. (**a**) Axial view of CT; (**b**) coronal view of CT.

## Data Availability

The data presented in this study are available on request from the corresponding author.

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
