# Peer review of "Transoral Marsupialization of an Isolated Surgical Ciliated Cyst of the Infratemporal Fossa"

_diagnostics, 2023, doi:10.3390/diagnostics13111825_

Round 1

Reviewer 1 Report

The authors present a case of an isolated surgical ciliated cyst that developed in a 40-year-old patient who was operated on for a mid-face fracture 25 years ago. The clinical symptoms and the results of the imaging studies are presented. They also report on possible alternatives that are important for differential diagnosis. As a surgical treatment, removal of the lesion was planned, for this reason a Le Fotr I osteotomy was performed. During surgery, it was revealed that it was probably a benign cyst, so instead of resecting, they performed marsupialization towards the sinus. In my opinion, the intervention should have been performed through the maxillary sinus, and intraoperative histology should have been performed. The lesion destroyed the posterior wall of the sinus, so this could have been a quick, simple intervention, the Penrose drain could have been inserted easily. Regardless, the article may be interesting and useful for readers. I recommend it for publication.

Author Response

Thank you for reviewing our manuscript and for providing us with your valuable comment.

Reviewer 2 Report

Highlighting the area of interest by drawing/ incorporating arrows or dotted circles in Figure 1 and Figure 3, will be appreciated.

  Authors can comment on the role of radiographic examination in the diagnosis of such cases.
The overall case report is interesting and useful to readers

Author Response

We appreciate the reviewer for valuable advice and detailed suggestions. Revised Figure 1. and 3. are submitted according to your comments.

Reviewer 3 Report

The present manuscript is well presented however the introduction section can be improved by adding some basic literature about the procedure for ready reference to readers. 

The minor spellchecks are required 

Author Response

Thank you for reviewing our manuscript and for providing us with your valuable comment. According to your minor spell checks were done.